# Video Diffusion Models

**Jonathan Ho**[*]
jonathanho@google.com

**Tim Salimans**[*]
salimans@google.com

**Alexey Gritsenko**
agritsenko@google.com

**William Chan**
williamchan@google.com

**Mohammad Norouzi**
mnorouzi@google.com

**David J. Fleet**
davidfleet@google.com

## Abstract

Generating temporally coherent high fidelity video is an important milestone in generative modeling research. We make progress towards this milestone by proposing a diffusion model for video generation that shows very promising initial results. Our model is a natural extension of the standard image diffusion architecture, and it enables jointly training from image and video data, which we find to reduce the variance of minibatch gradients and speed up optimization. To generate long and higher resolution videos we introduce a new conditional sampling technique for spatial and temporal video extension that performs better than previously proposed methods. We present the first results on a large text-conditioned video generation task, as well as state-of-the-art results on established benchmarks for video prediction and unconditional video generation. Supplementary material is available at https://video-diffusion.github.io/.

## 1 Introduction

Diffusion models have recently been producing high quality results in image generation and audio generation [e.g. 28, 39, 40, 16, 23, 36, 48, 60, 42, 10, 29], and there is significant interest in validating diffusion models in new data modalities. In this work, we present first results on video generation using diffusion models, for both unconditional and conditional settings.

We show that high quality videos can be generated using essentially the standard formulation of the Gaussian diffusion model [46], with little modification other than straightforward architectural changes to accommodate video data within the memory constraints of deep learning accelerators. We train models that generate a fixed number of video frames using a 3D U-Net diffusion model architecture, and we enable generating longer videos by applying this model autoregressively using a new method for conditional generation. We additionally show the benefits of joint training on video and image modeling objectives. We test our methods on video prediction and unconditional video generation, where we achieve state-of-the-art sample quality scores, and we also show promising first results on text-conditioned video generation.

## 2 Background

A diffusion model [46, 47, 22] specified in continuous time [53, 48, 10, 28] is a generative model with latents $\mathbf{z} = \{\mathbf{z}_t \mid t \in [0, 1]\}$ obeying a forward process $q(\mathbf{z}|\mathbf{x})$ starting at data $\mathbf{x} \sim p(\mathbf{x})$. The forward process is a Gaussian process that satisfies the Markovian structure:

$$q(\mathbf{z}_t|\mathbf{x}) = \mathcal{N}(\mathbf{z}_t; \alpha_t\mathbf{x}, \sigma_t^2\mathbf{I}), \quad q(\mathbf{z}_t|\mathbf{z}_s) = \mathcal{N}(\mathbf{z}_t; (\alpha_t/\alpha_s)\mathbf{z}_s, \sigma_{t|s}^2\mathbf{I}) \tag{1}$$

---

[*]Equal contribution

36th Conference on Neural Information Processing Systems (NeurIPS 2022).

where $0 \leq s < t \leq 1$, $\sigma_{t|s}^2 = (1 - e^{\lambda_t - \lambda_s})\sigma_t^2$, and $\alpha_t, \sigma_t$ specify a differentiable noise schedule whose log signal-to-noise-ratio $\lambda_t = \log[\alpha_t^2/\sigma_t^2]$ decreases with $t$ until $q(\mathbf{z}_1) \approx \mathcal{N}(\mathbf{0}, \mathbf{I})$.

**Training**  Learning to reverse the forward process for generation can be reduced to learning to denoise $\mathbf{z}_t \sim q(\mathbf{z}_t|\mathbf{x})$ into an estimate $\hat{\mathbf{x}}_\theta(\mathbf{z}_t, \lambda_t) \approx \mathbf{x}$ for all $t$ (we will drop the dependence on $\lambda_t$ to simplify notation). We train this denoising model $\hat{\mathbf{x}}_\theta$ using a weighted mean squared error loss

$$\mathbb{E}_{\boldsymbol{\epsilon}, t}\left[w(\lambda_t)\|\hat{\mathbf{x}}_\theta(\mathbf{z}_t) - \mathbf{x}\|_2^2\right] \tag{2}$$

over uniformly sampled times $t \in [0, 1]$. This reduction of generation to denoising can be justified as optimizing a weighted variational lower bound on the data log likelihood under the diffusion model, or as a form of denoising score matching [56, 47, 22, 28]. In practice, we use the $\boldsymbol{\epsilon}$-prediction parameterization, defined as $\hat{\mathbf{x}}_\theta(\mathbf{z}_t) = (\mathbf{z}_t - \sigma_t \boldsymbol{\epsilon}_\theta(\mathbf{z}_t))/\alpha_t$, and train $\boldsymbol{\epsilon}_\theta$ using a mean squared error in $\boldsymbol{\epsilon}$ space with $t$ sampled according to a cosine schedule [37]. This corresponds to a particular weighting $w(\lambda_t)$ for learning a scaled score estimate $\boldsymbol{\epsilon}_\theta(\mathbf{z}_t) \approx -\sigma_t \nabla_{\mathbf{z}_t} \log p(\mathbf{z}_t)$, where $p(\mathbf{z}_t)$ is the true density of $\mathbf{z}_t$ under $\mathbf{x} \sim p(\mathbf{x})$ [22, 28, 48]. We also train using the $\mathbf{v}$-prediction parameterization for certain models [42].

**Sampling**  We use a variety of diffusion model samplers in this work. One is the discrete time ancestral sampler [22] with sampling variances derived from lower and upper bounds on reverse process entropy [46, 22, 37]. To define this sampler, first note that the forward process can be described in reverse as $q(\mathbf{z}_s|\mathbf{z}_t, \mathbf{x}) = \mathcal{N}(\mathbf{z}_s; \tilde{\boldsymbol{\mu}}_{s|t}(\mathbf{z}_t, \mathbf{x}), \tilde{\sigma}_{s|t}^2 \mathbf{I})$ (noting $s < t$), where

$$\tilde{\boldsymbol{\mu}}_{s|t}(\mathbf{z}_t, \mathbf{x}) = e^{\lambda_t - \lambda_s}(\alpha_s/\alpha_t)\mathbf{z}_t + (1 - e^{\lambda_t - \lambda_s})\alpha_s \mathbf{x} \quad \text{and} \quad \tilde{\sigma}_{s|t}^2 = (1 - e^{\lambda_t - \lambda_s})\sigma_s^2. \tag{3}$$

Starting at $\mathbf{z}_1 \sim \mathcal{N}(\mathbf{0}, \mathbf{I})$, the ancestral sampler follows the rule

$$\mathbf{z}_s = \tilde{\boldsymbol{\mu}}_{s|t}(\mathbf{z}_t, \hat{\mathbf{x}}_\theta(\mathbf{z}_t)) + \sqrt{(\tilde{\sigma}_{s|t}^2)^{1-\gamma}(\sigma_{t|s}^2)^\gamma}\boldsymbol{\epsilon} \tag{4}$$

where $\boldsymbol{\epsilon}$ is standard Gaussian noise, $\gamma$ is a hyperparameter that controls the stochasticity of the sampler [37], and $s, t$ follow a uniformly spaced sequence from 1 to 0.

Another sampler, which we found especially effective with our new method for conditional generation (Section 3.1), is the predictor-corrector sampler [48]. Our version of this sampler alternates between the ancestral sampler step (4) and a Langevin correction step of the form

$$\mathbf{z}_s \leftarrow \mathbf{z}_s - \frac{1}{2}\delta\sigma_s \boldsymbol{\epsilon}_\theta(\mathbf{z}_s) + \sqrt{\delta}\sigma_s \boldsymbol{\epsilon}' \tag{5}$$

where $\delta$ is a step size which we fix to 0.1 here, and $\boldsymbol{\epsilon}'$ is another independent sample of standard Gaussian noise. The purpose of the Langevin step is to help the marginal distribution of each $\mathbf{z}_s$ generated by the sampler to match the true marginal under the forward process starting at $\mathbf{x} \sim p(\mathbf{x})$.

In the conditional generation setting, the data $\mathbf{x}$ is equipped with a conditioning signal $\mathbf{c}$, which may represent a class label, text caption, or other type of conditioning. To train a diffusion model to fit $p(\mathbf{x}|\mathbf{c})$, the only modification that needs to be made is to provide $\mathbf{c}$ to the model as $\hat{\mathbf{x}}_\theta(\mathbf{z}_t, \mathbf{c})$. Improvements to sample quality can be obtained in this setting by using *classifier-free guidance* [20]. This method samples using adjusted model predictions $\tilde{\boldsymbol{\epsilon}}_\theta$, constructed via

$$\tilde{\boldsymbol{\epsilon}}_\theta(\mathbf{z}_t, \mathbf{c}) = (1 + w)\boldsymbol{\epsilon}_\theta(\mathbf{z}_t, \mathbf{c}) - w\boldsymbol{\epsilon}_\theta(\mathbf{z}_t), \tag{6}$$

where $w$ is the *guidance strength*, $\boldsymbol{\epsilon}_\theta(\mathbf{z}_t, \mathbf{c}) = \frac{1}{\sigma_t}(\mathbf{z}_t - \hat{\mathbf{x}}_\theta(\mathbf{z}_t, \mathbf{c}))$ is the regular conditional model prediction, and $\boldsymbol{\epsilon}_\theta(\mathbf{z}_t)$ is a prediction from an unconditional model jointly trained with the conditional model (if $\mathbf{c}$ consists of embedding vectors, unconditional modeling can be represented as $\mathbf{c} = \mathbf{0}$). For $w > 0$ this adjustment has the effect of over-emphasizing the effect of conditioning on the signal $\mathbf{c}$, which tends to produce samples of lower diversity but higher quality compared to sampling from the regular conditional model [20]. The method can be interpreted as a way to guide the samples towards areas where an implicit classifier $p(\mathbf{c}|\mathbf{z}_t)$ has high likelihood, and is an adaptation of the explicit classifier guidance method proposed by [16].

# 3 Video diffusion models

Our approach to video generation using diffusion models is to use the standard diffusion model formalism described in Section 2 with a neural network architecture suitable for video data. Each of our models is trained to jointly model a fixed number of frames at a fixed spatial resolution. To extend sampling to longer sequences of frames or higher spatial resolutions, we will repurpose our models with a conditioning technique described later in Section 3.1.

In prior work on image modeling, the standard architecture for $\hat{\mathbf{x}}_\theta$ in an image diffusion model is a U-Net [38, 44], which is a neural network architecture constructed as a spatial downsampling pass followed by a spatial upsampling pass with skip connections to the downsampling pass activations. The network is built from layers of 2D convolutional residual blocks, for example in the style of the Wide ResNet [65], and each such convolutional block is followed by a spatial attention block [55, 58, 11]. Conditioning information, such as $\mathbf{c}$ and $\lambda_t$, is provided to the network in the form of an embedding vector added into each residual block (we find it helpful for our models to process these embedding vectors using several MLP layers before adding).

We propose to extend this image diffusion model architecture to video data, given by a block of a fixed number of frames, using a particular type of 3D U-Net [13] that is factorized over space and time. First, we modify the image model architecture by changing each 2D convolution into a space-only 3D convolution, for instance, we change each 3x3 convolution into a 1x3x3 convolution (the first axis indexes video frames, the second and third index the spatial height and width). The attention in each spatial attention block remains as attention over space; i.e., the first axis is treated as a batch axis. Second, after each spatial attention block, we insert a temporal attention block that performs attention over the first axis and treats the spatial axes as batch axes. We use relative position embeddings [45] in each temporal attention block so that the network can distinguish ordering of frames in a way that does not require an absolute notion of video time. We visualize the model architecture in Fig. 1.

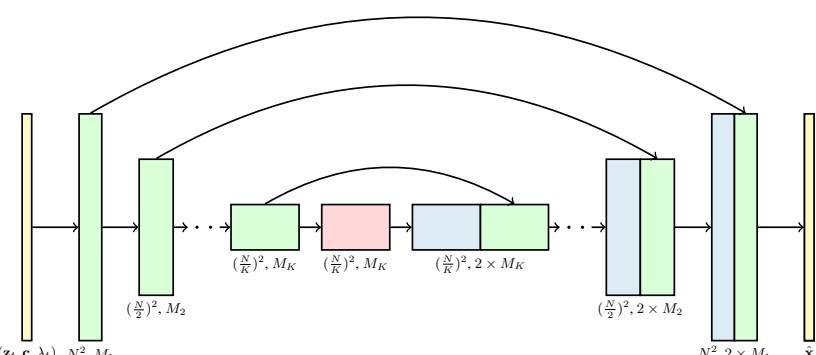

Figure 1: The 3D U-Net architecture for $\hat{\mathbf{x}}_\theta$ in the diffusion model. Each block represents a 4D tensor with axes labeled as frames $\times$ height $\times$ width $\times$ channels, processed in a space-time factorized manner as described in Section 3. The input is a noisy video $\mathbf{z}_t$, conditioning $\mathbf{c}$, and the log SNR $\lambda_t$. The downsampling/upsampling blocks adjust the spatial input resolution height $\times$ width by a factor of 2 through each of the $K$ blocks. The channel counts are specified using channel multipliers $M_1$, $M_2$, ..., $M_K$, and the upsampling pass has concatenation skip connections to the downsampling pass.

The use of factorized space-time attention is known to be a good choice in video transformers for its computational efficiency [2, 5, 21]. An advantage of our factorized space-time architecture, which is unique to our video generation setting, is that it is particularly straightforward to mask the model to run on independent images rather than a video, simply by removing the attention operation inside each time attention block and fixing the attention matrix to exactly match each key and query vector at each video timestep. The utility of doing so is that it allows us to jointly train the model on both video and image generation. We find in our experiments that this joint training is important for sample quality (Section 4).

## 3.1 Reconstruction-guided sampling for improved conditional generation

The videos we consider modeling typically consist of hundreds to thousands of frames, at a frame rate of at least 24 frames per second. To manage the computational requirements of training our models, we only train on a small subset of say 16 frames at a time. However, at test time we can generate longer videos by extending our samples. For example, we could first generate a video $\mathbf{x}^a \sim p_\theta(\mathbf{x})$ consisting of 16 frames, and then extend it with a second sample $\mathbf{x}^b \sim p_\theta(\mathbf{x}^b|\mathbf{x}^a)$. If $\mathbf{x}^b$ consists of frames following $\mathbf{x}^a$, this allows us to autoregressively extend our sampled videos to arbitrary lengths, which we demonstrate in Section 4.3.3. Alternatively, we could choose $\mathbf{x}^a$ to represent a video of lower frame rate, and then define $\mathbf{x}^b$ to be those frames in between the frames of $\mathbf{x}^a$. This allows one to then to upsample a video temporally, similar to how [34] generate high resolution images through spatial upsampling.

Both approaches require one to sample from a conditional model, $p_\theta(\mathbf{x}^b|\mathbf{x}^a)$. This conditional model could be trained explicitly, but it can also be derived approximately from our unconditional model $p_\theta(\mathbf{x})$ by imputation, which has the advantage of not requiring a separately trained model. For example, [48] present a general method for conditional sampling from a jointly trained diffusion model $p_\theta(\mathbf{x} = [\mathbf{x}^a, \mathbf{x}^b])$: In their approach to sampling from $p_\theta(\mathbf{x}^b|\mathbf{x}^a)$, the sampling procedure for updating $\mathbf{z}_s^b$ is unchanged from the standard method for sampling from $p_\theta(\mathbf{z}_s|\mathbf{z}_t)$, with $\mathbf{z}_s = [\mathbf{z}_s^a, \mathbf{z}_s^b]$, but the samples for $\mathbf{z}_s^a$ are replaced by exact samples from the forward process, $q(\mathbf{z}_s^a|\mathbf{x}^a)$, at each iteration. The samples $\mathbf{z}_s^a$ then have the correct marginal distribution by construction, and the samples $\mathbf{z}_s^b$ will conform with $\mathbf{z}_s^a$ through their effect on the denoising model $\hat{\mathbf{x}}_\theta([\mathbf{z}_t^a, \mathbf{z}_t^b])$. Similarly, we could sample $\mathbf{z}_s^a$ from $q(\mathbf{z}_s^a|\mathbf{x}^a, \mathbf{z}_t^a)$, which follows the correct conditional distribution in addition to the correct marginal. We will refer to both of these approaches as the *replacement* method for conditional sampling from diffusion models.

When we tried the replacement method to conditional sampling, we found it to not work well for our video models: Although samples $\mathbf{x}^b$ looked good in isolation, they were often not coherent with $\mathbf{x}^a$. This is caused by a fundamental problem with this replacement sampling method. That is, the latents $\mathbf{z}_s^b$ are updated in the direction provided by $\hat{\mathbf{x}}_\theta^b(\mathbf{z}_t) \approx \mathbb{E}_q[\mathbf{x}^b|\mathbf{z}_t]$, while what is needed instead is $\mathbb{E}_q[\mathbf{x}^b|\mathbf{z}_t, \mathbf{x}^a]$. Writing this in terms of the score of the data distribution, we get $\mathbb{E}_q[\mathbf{x}^b|\mathbf{z}_t, \mathbf{x}^a] = \mathbb{E}_q[\mathbf{x}^b|\mathbf{z}_t] + (\sigma_t^2/\alpha_t)\nabla_{\mathbf{z}_t^b} \log q(\mathbf{x}^a|\mathbf{z}_t)$, where the second term is missing in the replacement method. Assuming a perfect denoising model, plugging in this missing term would make conditional sampling exact. Since $q(\mathbf{x}^a|\mathbf{z}_t)$ is not available in closed form, however, we instead propose to approximate it using a Gaussian of the form $q(\mathbf{x}^a|\mathbf{z}_t) \approx \mathcal{N}[\hat{\mathbf{x}}_\theta^a(\mathbf{z}_t), (\sigma_t^2/\alpha_t^2)\mathbf{I}]$, where $\hat{\mathbf{x}}_\theta^a(\mathbf{z}_t)$ is a reconstruction of the conditioning data $\mathbf{x}^a$ provided by our denoising model. Assuming a perfect model, this approximation becomes exact as $t \to 0$, and empirically we find it to be good for larger $t$ also. Plugging in the approximation, and adding a weighting factor $w_r$, our proposed method to conditional sampling is a variant of the replacement method with an adjusted denoising model, $\tilde{\mathbf{x}}_\theta^b$, defined by

$$\tilde{\mathbf{x}}_\theta^b(\mathbf{z}_t) = \hat{\mathbf{x}}_\theta^b(\mathbf{z}_t) - \frac{w_r \alpha_t}{2}\nabla_{\mathbf{z}_t^b}\|\mathbf{x}^a - \hat{\mathbf{x}}_\theta^a(\mathbf{z}_t)\|_2^2 . \tag{7}$$

The additional gradient term in this expression can be interpreted as a form of *guidance* [16, 20] based on the model's reconstruction of the conditioning data, and we therefore refer to this method as *reconstruction-guided sampling*, or simply *reconstruction guidance*. Like with other forms of guidance, we find that choosing a larger weighting factor, $w_r > 1$, tends to improve sample quality. We empirically investigate reconstruction guidance in Section 4.3.3, where we find it to work surprisingly well, especially when combined with predictor-corrector samplers using Langevin diffusion [48].

Reconstruction guidance also extends to the case of spatial interpolation (or super-resolution), in which the mean squared error loss is imposed on a downsampled version of the model prediction, and backpropagation is performed through this downsampling. In this setting, we have low resolution ground truth videos $\mathbf{x}^a$ (e.g. at the 64x64 spatial resolution), which may be generated from a low resolution model, and we wish to upsample them into high resolution videos (e.g. at the 128x128 spatial resolution) using an unconditional high resolution diffusion model $\hat{\mathbf{x}}_\theta$. To accomplish this, we adjust the high resolution model as follows:

$$\tilde{\mathbf{x}}_\theta(\mathbf{z}_t) = \hat{\mathbf{x}}_\theta(\mathbf{z}_t) - \frac{w_r \alpha_t}{2}\nabla_{\mathbf{z}_t}\|\mathbf{x}^a - \hat{\mathbf{x}}_\theta^a(\mathbf{z}_t)\|_2^2 \tag{8}$$

where $\hat{\mathbf{x}}_\theta^a(\mathbf{z}_t)$ is our model's reconstruction of the low-resolution video from $\mathbf{z}_t$, which is obtained by downsampling the high resolution output of the model using a differentiable downsampling algorithm such as bilinear interpolation. Note that it is also possible to simultaneously condition on low resolution videos while autoregressively extending samples at the high resolution using the same reconstruction guidance method. In Fig. 2, we show samples of this approach for extending 16x64x64 low resolution samples at frameskip 4 to 64x128x128 samples at frameskip 1 using a 9x128x128 diffusion model.

## 4   Experiments

We report our results on video diffusion models for unconditional video generation (Section 4.1), conditional video generation (video prediction) (Section 4.2), and text-conditioned video generation (Section 4.3). We evaluate our models using standard metrics such as FVD [54], FID [19], and IS [43]; details on evaluation are provided below alongside each benchmark. Samples and additional results are provided at `https://video-diffusion.github.io/`. Architecture hyperparameters, training details, and compute resources are listed in Appendix A.

### 4.1   Unconditional video modeling

To demonstrate our approach on unconditional generation, we use a popular benchmark of Soomro et al. [49] for unconditional modeling of video. The benchmark consists of short clips of people performing one of 101 activities, and was originally collected for the purpose of training action recognition models. We model short segments of 16 frames from this dataset, downsampled to a spatial resolution of 64x64. In Table 1 we present perceptual quality scores for videos generated by our model, and we compare against methods from the literature, finding that our method strongly improves upon the previous state-of-the-art.

We use the data loader provided by TensorFlow Datasets [1] without further processing, and we train on all 13,320 videos. Similar to previous methods, we use the C3D network [51][2] for calculating FID and IS, using 10,000 samples generated from our model. C3D internally resizes input data to the 112x112 spatial resolution, so perceptual scores are approximately comparable even when the data is sampled at a different resolution originally. As discussed by [64], methods in the literature are unfortunately not always consistent in the data preprocessing that is used, which may lead to small differences in reported scores between papers. The Inception Score we calculate for real data ($\approx 60$) is consistent with that reported by [26], who also report a higher real data Inception score of $\approx 90$ for data sampled at the 128x128 resolution, which indicates that our 64x64 model might be at a disadvantage compared to works that generate at a higher resolution. Nevertheless, our model obtains the best perceptual quality metrics that we could find in the literature.

| Method | Resolution | FID↓ | IS↑ |
|---|---|---|---|
| MoCoGAN [52] | 16x64x64 | $26998 \pm 33$ | 12.42 |
| TGAN-F [26] | 16x64x64 | $8942.63 \pm 3.72$ | 13.62 |
| TGAN-ODE [18] | 16x64x64 | $26512 \pm 27$ | 15.2 |
| TGAN-F [26] | 16x128x128 | $7817 \pm 10$ | $22.91 \pm .19$ |
| VideoGPT [62] | 16x128x128 | | $24.69 \pm 0.30$ |
| TGAN-v2 [41] | 16x64x64 | $3431 \pm 19$ | $26.60 \pm 0.47$ |
| TGAN-v2 [41] | 16x128x128 | $3497 \pm 26$ | $28.87 \pm 0.47$ |
| DVD-GAN [14] | 16x128x128 | | $32.97 \pm 1.7$ |
| **Video Diffusion (ours)** | 16x64x64 | $\mathbf{295 \pm 3}$ | $\mathbf{57 \pm 0.62}$ |
| real data | 16x64x64 | | 60.2 |

Table 1: Unconditional video modeling results on UCF101.

---

[2] We use the C3D model as implemented at `github.com/pfnet-research/tgan2` [41].

## 4.2 Video prediction

A common benchmark task for evaluating generative models of video is *video prediction*, where the model is given the first frame(s) of a video and is asked to generate the remainder. Models that do well on this *conditional generation* task are usually trained explicitly for this conditional setting, for example by being autoregressive across frames. Although our models are instead only trained unconditionally, we can adapt them to the video prediction setting by using the guidance method proposed in section 3.1. Here we evaluate this method on two popular video prediction benchmarks, obtaining state-of-the-art results.

**BAIR Robot Pushing** We evaluate video prediction performance on BAIR Robot Pushing [17], a standard benchmark in the video literature consisting of approximately 44000 videos of robot pushing motions at the 64x64 spatial resolution. Methods for this benchmark are conditioned on 1 frame and generate the next 15. Results are listed in Table 2. Following the evaluation protocol of [4] and others, we calculate FVD [54] using the I3D network [8] by comparing $100 \times 256$ model samples against the 256 examples in the evaluation set.

**Kinetics-600** We additionally evaluate video prediction performance on the Kinetics-600 benchmark [27, 9]. Kinetics-600 contains approximately 400 thousand training videos depicting 600 different activities. We train unconditional models on this dataset at the $64 \times 64$ resolution and evaluate on 50 thousand randomly sampled videos from the test set, where we condition on a randomly sampled subsequence of 5 frames and generate the next 11 frames. Like previous works, we calculate FVD and Inception Score using the I3D network [8]. See Table 3 for results. In our reported results we sample test videos without replacement, and we use the same randomly selected subsequences for generating model samples and for defining the ground truth, since this results in the lowest bias and variance in the reported FVD metric. However, from personal communication we learned that [33, 14] instead sampled *with replacement*, and used a different random seed when sampling the ground truth data. We find that this way of evaluating raises the FVD obtained by our model slightly, from 16.2 to 16.9. Inception Score is unaffected.

Table 2: Video prediction on BAIR Robot Pushing.

| Method | FVD↓ |
|---|---|
| DVD-GAN [14] | 109.8 |
| VideoGPT [62] | 103.3 |
| TrIVD-GAN-FP [33] | 103.3 |
| Transframer [35] | 100 |
| CCVS [31] | 99 |
| VideoTransformer [59] | 94 |
| FitVid [4] | 93.6 |
| NUWA [61] | 86.9 |
| Video Diffusion (ours) ancestral sampler, 512 steps | 68.19 |
| Langevin sampler, 256 steps | **66.92** |

Table 3: Video prediction on Kinetics-600.

| Method | FVD↓ | IS↑ |
|---|---|---|
| Video Transformer [59] | $170 \pm 5$ | |
| DVD-GAN-FP [14] | $69.1 \pm 0.78$ | |
| Video VQ-VAE [57] | $64.3 \pm 2.04$ | |
| CCVS [31] | $55 \pm 1$ | |
| TrIVD-GAN-FP [33] | $25.74 \pm 0.66$ | 12.54 |
| Transframer [35] | 25.4 | |
| Video Diffusion (ours) ancestral, 256 steps | 18.6 | 15.39 |
| Langevin, 128 steps | $\mathbf{16.2 \pm 0.34}$ | **15.64** |

## 4.3 Text-conditioned video generation

The remaining experiments reported are on text-conditioned video generation. In this text-conditioned video generation setting, we employ a dataset of 10 million captioned videos, and we condition the diffusion model on captions in the form of BERT-large embeddings [15] processed using attention pooling. We consider two model sizes: a small model for the joint training ablation, and a large model for generating the remaining results (both architectures are described in detail in Appendix A), and we explore the effects of joint video-image training, classifier-free guidance, and our newly proposed reconstruction guidance method for autoregressive extension and simultaneous spatial and temporal super-resolution. We report the following metrics in this section on 4096 samples: the video metric FVD, and the Inception-based image metrics FID and IS measured by averaging activations across frames (FID/IS-avg) and by measuring the first frame only (FID/IS-first). For FID and FVD, we report two numbers which are measured against the training and validation sets, respectively. For IS, we report two numbers which are averaged scores across 1 split and 10 splits of samples, respectively.

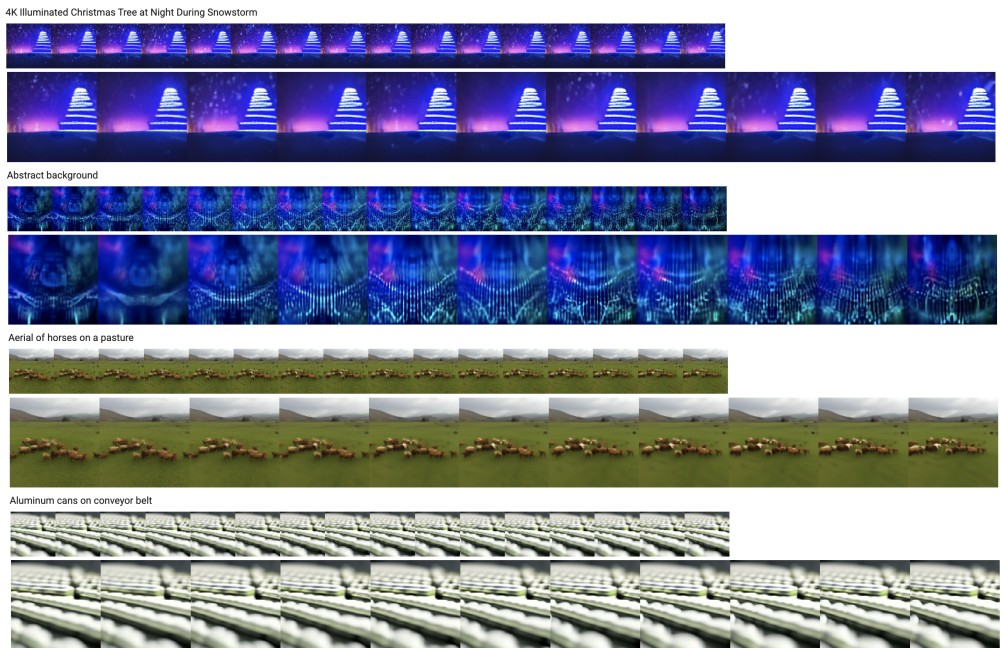

Figure 2: Text-conditioned video samples from a cascade of two models. First samples are generated from a 16x64x64 frameskip 4 model. Then those samples are treated as ground truth for simultaneous super-resolution and autoregressive extension to 64x128x128 using a 9x128x128 frameskip 1 model. Both models are conditioned on the text prompt. In this figure, the text prompt, low resolution frames, and high resolution frames are visualized in sequence. See Fig. 5 for more samples.

### 4.3.1 Joint training on video and image modeling

As described in Section 3, one of the main advantages of our video architecture is that it allows us to easily train the model jointly on video and image generative modeling objectives. To implement this joint training, we concatenate random independent image frames to the end of each video sampled from the dataset, and we mask the attention in the temporal attention blocks to prevent mixing information across video frames and each individual image frame. We choose these random independent images from random videos within the same dataset; in future work we plan to explore the effect of choosing images from other larger image-only datasets.

Table 4 reports results for an experiment on text-conditioned 16x64x64 videos, where we consider training on an additional 0, 4, or 8 independent image frames per video. One can see clear improvements in video and image sample quality metrics as more independent image frames are added. Adding independent image frames has the effect of reducing variance of the gradient at the expense of some bias for the video modeling objective, and thus it can be seen as a memory optimization to fit more independent examples in a batch.

Table 4: Improved sample quality due to image-video joint training on text-to-video generation.

| Image frames | FVD↓ | FID-avg↓ | IS-avg↑ | FID-first↓ | IS-first↑ |
|---|---|---|---|---|---|
| 0 | 202.28/205.42 | 37.52/37.40 | 7.91/7.58 | 41.14/40.87 | 9.23/8.74 |
| 4 | 68.11/70.74 | 18.62/18.42 | 9.02/8.53 | 22.54/22.19 | 10.58/9.91 |
| 8 | 57.84/60.72 | 15.57/15.44 | 9.32/8.82 | 19.25/18.98 | 10.81/10.12 |

### 4.3.2 Effect of classifier-free guidance

Table 5 reports results that verify the effectiveness of classifier-free guidance [20] on text-to-video generation. As expected, there is clear improvement in the Inception Score-like metrics with higher guidance weight, while the FID-like metrics improve and then degrade with increasing guidance weight. Similar findings have been reported on text-to-image generation [36].

Figure 3 shows the effect of classifier-free guidance [20] on a text-conditioned video model. Similar to what was observed in other work that used classifier-free guidance on text-conditioned image generation [36] and class-conditioned image generation [20, 16], adding guidance increases the sample fidelity of each individual image and emphases the effect of the conditioning signal.

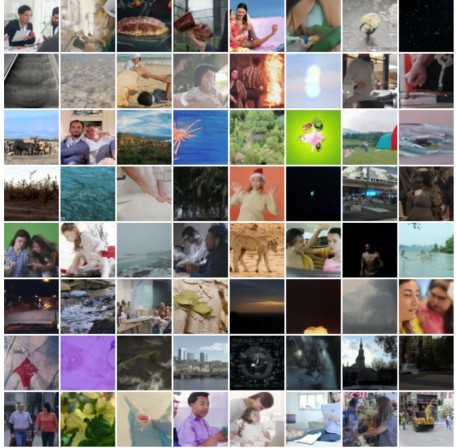 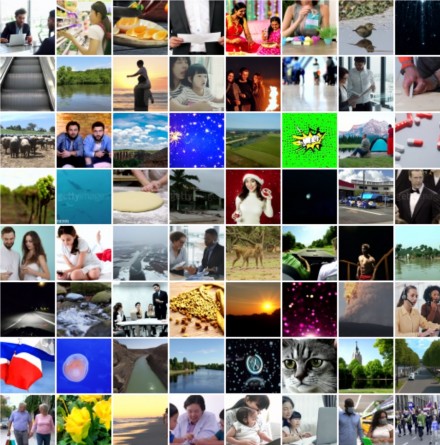

Figure 3: Example frames from a random selection of videos generated by our 16x64x64 text-conditioned model. Left: unguided samples, right: guided samples using classifier-free guidance.

Table 5: Effect of classifier-free guidance on text-to-video generation (large models). Sample quality is reported for 16x64x64 models trained on frameskip 1 and 4 data. The model was jointly trained on 8 independent image frames per 16-frame video.

| Frameskip | Guidance weight | FVD↓ | FID-avg↓ | IS-avg↑ | FID-first↓ | IS-first↑ |
|---|---|---|---|---|---|---|
| 1 | 1.0 | 41.65/43.70 | 12.49/12.39 | 10.80/10.07 | 16.42/16.19 | 12.17/11.22 |
| | 2.0 | 50.19/48.79 | 10.53/10.47 | 13.22/12.10 | 13.91/13.75 | 14.81/13.46 |
| | 5.0 | 163.74/160.21 | 13.54/13.52 | 14.80/13.46 | 17.07/16.95 | 16.40/14.75 |
| 4 | 1.0 | 56.71/60.30 | 11.03/10.93 | 9.40/8.90 | 16.21/15.96 | 11.39/10.61 |
| | 2.0 | 54.28/51.95 | 9.39/9.36 | 11.53/10.75 | 14.21/14.04 | 13.81/12.63 |
| | 5.0 | 185.89/176.82 | 11.82/11.78 | 13.73/12.59 | 16.59/16.44 | 16.24/14.62 |

### 4.3.3 Autoregressive video extension for longer sequences

In Section 3.1 we proposed the *reconstruction guidance method* for conditional sampling from diffusion models, an improvement over the *replacement method* of [48]. In Table 6 we present results on generating longer videos using both techniques, and find that our proposed method indeed improves over the replacement method in terms of perceptual quality scores.

Figure 4 shows the samples of our reconstruction guidance method for conditional sampling compared to the replacement method (Section 3.1) for the purposes of generating long samples in a block-autoregressive manner (Section 4.3.3). The samples from the replacement method clearly show a lack of temporal coherence, since frames from different blocks throughout the generated videos appear to be uncorrelated samples (conditioned on **c**). The samples from the reconstruction guidance method, by contrast, are clearly temporally coherent over the course of the entire autoregressive generation process. Figure 2 additionally shows samples of using the reconstruction guidance method to simultaneously condition on low frequency, low resolution videos while autoregressively extending temporally at a high resolution.

Table 6: Generating 64x64x64 videos using autoregressive extension of 16x64x64 models.

| Guidance weight | Conditioning method | FVD↓ | FID-avg↓ | IS-avg↑ | FID-first↓ | IS-first↑ |
|---|---|---|---|---|---|---|
| 2.0 | reconstruction guidance | 136.22/134.55 | 13.77/13.62 | 10.30/9.66 | 16.34/16.46 | 14.67/13.37 |
| | replacement | 451.45/436.16 | 25.95/25.52 | 7.00/6.75 | 16.33/16.46 | 14.67/13.34 |
| 5.0 | reconstruction guidance | 133.92/133.04 | 13.59/13.58 | 10.31/9.65 | 16.28/16.53 | 15.09/13.72 |
| | replacement | 456.24/441.93 | 26.05/25.69 | 7.04/6.78 | 16.30/16.54 | 15.11/13.69 |

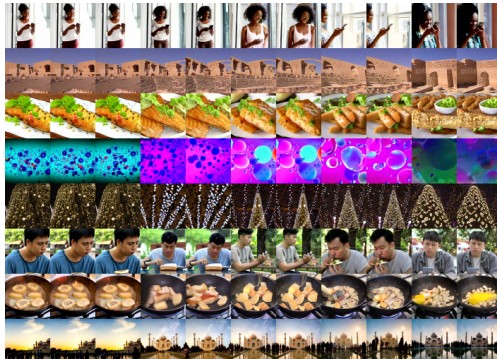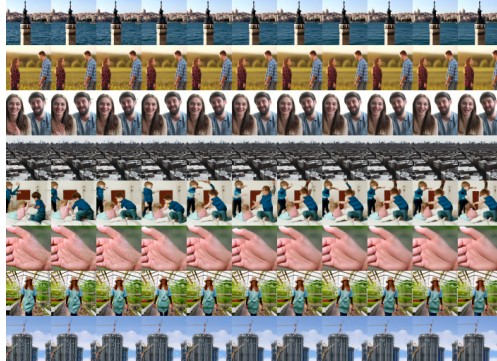

Figure 4: Comparing the replacement method (left) vs the reconstruction guidance method (right) for conditioning for block-autoregressive generation of 64 frames from a 16 frame model. Video frames are displayed over time from left to right; each row is an independent sample. The replacement method suffers from a lack of temporal coherence, unlike the reconstruction guidance method.

## 5 Related work

Prior work on video generation has usually employed other types of generative models, notably, autoregressive models, VAEs, GANs, and normalizing flows [e.g. 3, 4, 32, 30, 14, 59, 62, 57]. Related work on model classes similar to diffusion models includes [25, 24]. Concurrent work [63] proposes a diffusion-based approach to video generation that uses an image diffusion model to predict each individual frame within a RNN temporal autoregressive model. Our video diffusion model, by contrast, jointly models entire videos (blocks of frames) using a 3D video architecture with interleaved spatial and temporal attention, and we extend to long sequence lengths by filling in frames or autoregressive temporal extension.

## 6 Conclusion

We have introduced diffusion models for video modeling, thus bringing recent advances in generative modeling using diffusion models to the video domain. We have shown that with straightforward extensions of conventional U-Net architectures for 2D image modeling to 3D space-time, with factorized space-time attention blocks, one can learn effective generative models for video data using the standard formulation of the diffusion model. This includes unconditional models, text-conditioned models, and video prediction models.

We have additionally demonstrated the benefits of joint image-video training and classifier-free guidance for video diffusion models on both video and image sample quality metrics, and we also introduced a new reconstruction-guided conditional sampling method that outperforms existing replacement or imputation methods for conditional sampling from unconditionally trained models. Our reconstruction guidance method can generate long sequences using either frame interpolation (or temporal super-resolution) or extrapolation in an auto-regressive fashion, and also can perform spatial super-resolution. We look forward to investigating this method in a wider variety of conditioning settings.

Our goal with this work is to advance research on methods in generative modeling, and our methods have the potential to positively impact creative downstream applications. As with prior work in generative modeling, however, our methods have the potential for causing harmful impact and could enhance malicious or unethical uses of generative models, such as fake content generation, harassment, and misinformation spread, and thus we have decided not to release our models. Like all generative models, our models reflect the biases of their training datasets and thus may require curation to ensure fair results from sampling. In particular, our text-to-video models inherit the challenges faced by prior work on text-to-image models, and our future work will involve auditing for forms of social bias, similar to [6, 7, 50, 12] for image-to-text and image labeling models. We see our work as only a starting point for further investigation on video diffusion models and investigation into their societal implications, and we will aim to explore benchmark evaluations for social and cultural bias in the video generation setting and make the necessary research advances to address them.

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
