# OpenReview forum: "Video Diffusion Models"
_NeurIPS.cc/2022/Conference — NeurIPS 2022 Accept_

### Official Review · Reviewer_EcMx · 2022-07-10

**Rating:** 7
**Confidence:** 5
**Soundness:** 4 excellent
**Presentation:** 4 excellent
**Contribution:** 4 excellent

**Summary:**

The paper proposed new diffusion model for video generation. The proposed model use 3D-Unet as a backbone for the diffusion model. Furthermore, the paper introduce novel conditional sampling technique for spatial and temporal video generation. The proposed methods present on unconditional video modeling, video prediction, and text conditioned video generation. Moreover, the proposed method show large improvement comparing to the baselines.

**Questions:**

It will be helpful to provide the reason why you choose to use the 3D-Unet. Are there any other architecture that can be also appropriate for this task?

Regarding the dataset that chosen - Can you think about dataset that has more diverse content and interactions? for example, not only video that has repetitive movements.

**Limitations:**

The author provide an section regarding potential negative societal impact of their work.

**Strengths And Weaknesses:**

Strengths:
- The paper introduce a novel diffusion model for video generation
- The paper proposed two novel ideas to drive video X^b from sampled video X^a
- The improvement in the results is relatively high.
- The paper is well written and easy to follow.



Weaknesses:
- Results - looking on the generated video, the resolution is low. Can you trained the model for higher resolution? We all know that diffusion model is capable to generate high fidelity images, so it seems straight forward to use the super resolution approach like in CDM [cascade diffusion model] and get better video.
- The paper should compare to other previous results. for example, StyleGAN-V [CVPR22], DIGAN [ICLR22].

---

> ### Author Response · Authors · 2022-08-02
> **Response**
>
> Thank you for your review. We address your questions and concerns below:
>
> 1. **Low resolution**
>
>     Since video generation is computationally expensive, and since most well established benchmarks in the video generation literature only consider low resolution, we chose to focus on this setting for this paper. However, we will add a higher resolution (128x128) experiment for the UCF101 benchmark where there are some other works to compare against.
>
> 2. **Comparing against recent works StyleGAN-V [CVPR22], DIGAN [ICLR22]**
>
>     Thanks for making us aware of these. We will add them to the discussion and include their results on our considered benchmarks (only UCF101 it seems).
>
> 3. **Why 3D-Unet**
>
>     We experimented with models that used 3D convolution, or separable convolution across space and time. However, we found these models to not be an improvement over the separable space-time attention + spatial convolution that we ended up using. We also tried feedforward models outside the UNet family, such as ResNets and ViTs, but found their performance to be much worse than our UNet. There might well exist other architectures that do well on this task, but we have not identified them as of yet.
>
> 4. **About diverse datasets**
>
>     Of the publicly available datasets it seems that Kinetics is the most diverse, containing videos of people performing 600 different actions, which is one of the reasons we experimented on this dataset in particular. The private dataset we consider for text-conditional video generation is even more diverse, also including many scenes without people in them. Unfortunately the latter cannot be made public. Do you know any other diverse public datasets that we could consider?

---

### Official Review · Reviewer_xwAd · 2022-07-12

**Rating:** 9
**Confidence:** 5
**Soundness:** 4 excellent
**Presentation:** 4 excellent
**Contribution:** 4 excellent

**Summary:**

This paper adapts diffusion models for image generation to generate temporally coherent videos. The adaptation were done by (1) using a 3D U-Net diffusion model architecture to generate a fixed number of video frames; (2) using reconstruction guidance to generate temporally coherent longer videos. This paper claims that the proposed method generates SOTA results on benchmarks for unconditioned video generation and video prediction. The paper also claims that joint training using both images and videos improves the generated video quality.

**Questions:**

Given the training hardware used, how long does training take on these tasks?

**Limitations:**

Yes.

**Strengths And Weaknesses:**

Strength:
- The approach is a natural retention from diffusion models for image generation.
- Results are great both in terms of visual quality and numerical results.

Weaknesses:
- The claim of joint training using images and videos is better supported with more experiments:
1. As mentioned in the paper, images are better sampled outside of the video dataset.
2. In the unconditional video generation task, do joint training help?
- Video frame interpolation related literatures are better discussed in the sections of the video extension.

---

> ### Author Response · Authors · 2022-08-02
> **Response**
>
>
> Thank you for your review and your appreciation of our work. We briefly address the weaknesses you identified below:
>
> 1. **More experiments needed for supporting the joint training using images and video**
>
>     a. **It would be better to sample images outside of the video dataset**
>
>     Yes, adding a more diverse set of images to our training data would help further improve our results. We hope to show this in future work, as it’s not part of the benchmarks / datasets we consider here. Our goal in this paper is to study joint training on images for a single video benchmark dataset without introducing confounding factors due to images from extra datasets.
>
>     b. **Does joint training work for unconditional video generation?**
>
>     Yes, we find that joint training also helps unconditional video modeling reach better results, with the advantage being greatest when training on smaller minibatches / larger datasets. We could add an additional ablation for this, e.g. by running UCF101 with and without joint training. Would that address your concern or are you looking for something different?
>
> 2. **Video frame interpolation literature would be better discussed in the section on video extension**
>
>     We will review our discussion of this literature and move the content to the video extension section where appropriate.

---

### Official Review · Reviewer_m1dT · 2022-07-12

**Rating:** 6
**Confidence:** 2
**Soundness:** 3 good
**Presentation:** 3 good
**Contribution:** 3 good

**Summary:**

This paper shows that diffusion models also perform well for video modeling. By choosing a factorized attention module, it can directly extend a 2D U-Net to 3D space-time, allowing joint training with images possible. Results on unconditional text-to-video generation and video prediction show the strong performance of the proposed model.



**Questions:**

1. The major technical contribution seems to come from [39] and the novelty on videos such as factorized attention and co-train are extensively studied by previous works. Nevertheless, combining these components together are worth noting.

2. The writing is more or less like a draft version, it makes people with less prior knowledge very hard to follow.

3. In formula (8), How much extra training time does the reconstruction cost?

**Limitations:**

Yes, the authors adequately addressed the limitations and potential negative societal impact.

**Strengths And Weaknesses:**

1. A straightforward extension of [39] to videos with a factorized encoder, thus allowing joint training with images possible. It shows tricks like classifier-free guidance is also effective in video generation.

2. Ablate different sampling methods in Table 6 shows that the proposed reconstruction-guided conditional sampling method performs the best.

3. Results on UCF101, K600, and BAIR-RP in Table 1, 2, 3 are outstanding.

---

> ### Author Response · Authors · 2022-08-02
> **Response**
>
> Thank you for your review. We briefly address your questions below:
>
> 1. **Novelty**
>
>     Joint-training between images and video is not something we have seen before for the class of diffusion models. We will gladly add a reference if provided. We would also like to highlight our proposed guidance method for conditional sampling from unconditional models, which produces great results and seems quite different from earlier methods.
>
> 2. **Writing**
>
>     We have made some edits for the next version, and we can add more background material to make the paper more accessible to a wider audience. Is there any part of the writing specifically that we should have another look at before submitting the camera ready version?
>
> 3. **Training time for reconstruction guidance**
>
>     Our guidance technique for conditional sampling of unconditionally trained models is only applied at sampling time: Training is unconditional, so no additional training time is required.

---

> > ### Comment · Reviewer_m1dT · 2022-08-08
> > **Thanks for the response**
> >
> > Thanks for the response!
> > 1. I believe co-train/co-finetune is well studied in image and video recognition domain so directly extending it into video generation sounds a bit weak to me. Though, the guidance method sounds interesting and thanks the author[s] for the contribution.
> > 2. Yes, a more comprehensive background recap will be much more appreciated, how about adding it into Appendix?
> > 3. Thanks for the answer.
> >
> > I have updated my scores accordingly, Thanks.

---

### Official Review · Reviewer_hVqw · 2022-07-12

**Rating:** 5
**Confidence:** 3
**Soundness:** 3 good
**Presentation:** 3 good
**Contribution:** 3 good

**Summary:**

The paper extends the image-based diffusion model methods into the video domain. It builds upon the 3D U-Net architecture with attentions along the the decoupled space and time dimensions. The proposed video diffusion model also enables text-conditioned video generation. They outperform state-of-the-art methods in different benchmarks for video prediction and unconditional video generation. They also show the joint image-video training and classifier-free guidance for video diffusion models. The proposed diffusion model can also perform on the long sequence via auto-regressive inference.


**Questions:**

* Is the space-time factorization necessary? What would be the performance vs efficiency of using 3D convolution based U-Net compared to the proposed factorized one?
* How much performance drop will happen without the temporal attention, ie, per-frame method?

**Limitations:**

The authors have adequately addressed the limitations and potential negative societal impact of their work.

**Strengths And Weaknesses:**

Strengths
1. The proposed 3D U-Net diffusion model achieves state-of-the-art performances on unconditional video modeling on UCF101, video prediction on BAIR Robot Pushing and Kinetics-600, and text-based video generation tasks.
2. The proposed joint image and video training is effective.
3. The paper starts with the background section which gives a good guidance to readers.
----

Weaknesses
1. How does the proposed method work on 128x128 resolution results, compared to those of 64x64?
2. In figure 1, where is the num-frames dimension denoted?
3. Is the space-time factorization necessary? What would be the performance vs efficiency of using 3D convolution based U-Net compared to the proposed factorized one?
4. How much performance drop will happen without the temporal attention, ie, per-frame method?

---

> ### Author Response · Authors · 2022-08-02
> **Response**
>
> Thank you for your review. We briefly address your 4 identified weaknesses below:
>
> 1. **128x128 vs 64x64 resolution**
>
>     Working with video data is more computationally demanding than working with images. For this reason most of the popular benchmarks only consider low resolution. For UCF101 there also exist some works considering 128x128, as we discuss in the paper. For this benchmark, evaluation metrics are comparable between 64x64 and 128x128, since the C3D network used for evaluation does an internal resizing. (Simply resizing our 64x64 samples to 128x128 before evaluation will therefore not affect the internal representation of our samples). We’re therefore confident the method will work at that resolution also. If it addresses your concern, we can add an experiment at the 128x128 resolution for this benchmark.
>
> 2. **Figure 1 dimensions**
>
>     Figure 1 shows how our 4-dimensional video tensors are downsampled and upsampled in the spatial dimensions. Since the frame dimension is not downsampled, we do not denote the number of frames in this figure. We can make this explicit in the caption of the figure.
>
> 3. **Necessity of space-time factorization**
>
>     Space-time factorization is indeed essential in keeping memory requirements for parameters reasonable. In addition, a space-time factorized architecture more easily admits joint training on videos and images, which we found essential for good sample quality. This is because the factorized architecture lets us easily drop or bypass the layers that perform temporal mixing when training on independent images, and these layers have relatively few parameters. Performing an analogous dropping of parameters for 3D convolutions to bypass temporal mixing would waste many more parameters.
>
> 4. **Performance without temporal attention**
>
>     Without temporal attention all frames of the video would be generated independently. This is effectively how we model images in our joint training of video and image generation. The generated frames would still look good in this case, and the metrics that look at single frames (“FID-first” and “IS-first” in table 4) would not be affected. However, all metrics that look at the relationship between frames (all other reported metrics) would be much worse. As a baseline we tested outputting the same frame for the entire video for the video prediction benchmarks (BAIR and Kinetics) and found that the results are indeed bad. Outputting independent frames is presumably even worse on these metrics.

---

### Meta-Review · Area_Chair_oqYi · 2022-08-23

**Recommendation:** Accept
**Confidence:** Certain

**Metareview:**

This paper proposes a diffusion model for video capable of generating long and high-resolution videos. Diffusion models have generated some more excitement around generative models as well, so the paper is well-timed. The reviewers had a few concerns regarding additional experiments and clarifications, and it appears that the authors have satisfied those concerns. Overall, the reviews are positive, and there was a decent amount of interaction between the reviewers and authors, though much of the discussion was straightforward and didn't seem to require a good deal of discussion.

I therefore recommend accept for the paper based on there being a clear consensus and the discussions and revision satisfying most outstanding concerns.

Overall I have no recommendations of the reviewers based on this paper. The paper might have been very straightforward to read, and given the consensus I think that it is the case that this was a relatively easy review for everyone (neural reviewer score for everyone, though maybe due to paper being easy).

**Award:**

No

---

### Decision · Program_Chairs · 2022-09-14

Accept